# Imagining the Latent Space of a Variational Auto-Encoders

## Abstract

Variational Auto-Encoders (VAEs) are designed to capture compressible information about a dataset. As a consequence the information stored in the latent space is seldom sufficient to reconstruct a particular image. To help understand the type of information stored in the latent space we train a GAN-style decoder constrained to produce images that the VAE encoder will map to the same region of latent space. This allows us to "imagine" the information captured in the latent space. We argue that this is necessary to make a VAE into a truly generative model. We use our GAN to visualise the latent space of a standard VAE and of a $\beta$-VAE.

## 1 Introduction

Variational auto-encoders (VAEs) have made a significant impact since their introduction by Kingma and Welling (2014). However, one of their perceived problems is their reconstruction performance. This has spawned a wave of research into trying to improve the reconstruction performance (Zhao et al., 2017; Dai and Wipf, 2019; Larsen et al., 2016; Gao et al., 2017; Brock et al., 2017). We argue that such attempts are misguided. The whole point of VAEs is to capture only compressible information and discard information specific to any particular image. This is a consequence of the well known *evidence lower bound* or ELBO objective function consisting of a negative log-probability of generating the original image from the latent representation (this is often implemented as a mean squared error between the image and the reconstruction, although as we argue in Appendix A this term should be proportional to the logarithm of the mean squared error) and a KL-divergence between the probability distribution representing a latent code and a 'prior distribution' (usually taken as a multivariate normal with mean zero and unit variance). These two terms have a nice interpretation in terms of the *minimum description length* (Rissanen, 1978)—this has been described elsewhere, for example, Chen et al. (2016). The KL-term can be viewed as a measure of the amount of information in the latent code while the log-probability of the image measures the amount of information required to change the image produced by the decoder into the input image (see Section 3 for details). That is, the latent space of a VAE can be viewed as a model of the dataset—capturing compressible information while not encoding any image specific information (which is cheaper to communicate using the reconstruction loss).

The great strength of a VAE is that it builds a model of the dataset that does not over-fit (i.e. code for in-compressible features found in specific images). However, because of this it typically will not do a good job of reconstructing images as the latent code does not contain enough information to do the reconstruction (for very restrictive dataset such as MNIST and Celeb-A a lot of information can be captured in the latent space, but for more complex datasets like ImageNet or CIFAR the reconstructions are poor). Of course, if you want good reconstructions on the training set then the simplest solution is to remove the KL-divergence term and just use an autoencoder. However, having a model that does not over-fit the dataset can be useful, but in this case the decoder of a standard VAE should *not* be regarded as a generative model—that is not its purpose. If we wish to generate realistic looking images we need to imagine the information discarded by the encoder. As a rather simplified analogy, consider a verbal description of an image "a five year old girl in a blue dress standing on a beach". If we asked different artists to depict such scene there is clearly not enough information to provide pixel-wise or feature-wise similarity between their interpretation although each artist could render a convincing image that satisfies the description. In a similar manner if we

want a VAE to act as a generative model we need to build a renderer that will imagine an image consistent with the latent variable representation.

A simple way to achieve this is using a modified Generative Adversarial Network (GAN). We call such a model a *latent space renderer*-GAN (or LSR-GAN). To generate an image we choose a latent vector $z$ from the prior distribution for the VAE. This is passed to a generator network that generates an image, $\hat{x}$, with the same dimensions as that of the dataset used to train the VAE. The generated image has both to convince a discriminator network that it is a real image—as is usual for a GAN (Goodfellow et al., 2014)—at the same time the VAE encoder should map $\hat{x}$ close to $z$. To accomplish this we add an additional cost to the normal GAN loss function for the generator ($L_{\text{GEN}}$)

$$L_{\text{GEN}} - \lambda \log(q_\phi(z|\hat{x})) \qquad (1)$$

where $q_\phi(\cdot|\hat{x})$ is the probability distribution generated by the VAE encoder given an image $\hat{x}$ and $z$ is the latent vector that was put into the GAN generator. Note that when training the LSR-GAN we freeze the weights of the VAE encoder. The constant $\lambda$ is an adjustable hyperparameter providing a trade-off between how realistic the image should look and how closely it captures the information in the latent space. This modification of the objective function can clearly be applied to any GAN or used with any VAE. Although the idea is simple, it provides a powerful method for visualising (imagining) the information stored in a latent space. Interestingly, it also appears to provide a powerful regularisation mechanism to stabilize the training for GANs.

Combinations of VAEs and GANs are, of course, not new (Makhzani et al., 2016; Larsen et al., 2016; Brock et al., 2017; Huang et al., 2018; Srivastava et al., 2017). In all cases we are aware of GANs have been combined with VAEs to "correct" for the poor reconstruction performance of the VAE (see Appendix B for a more detailed discussion of the literature on VAE-GAN hybrids). As we have argued (and expound on in more detail in Section 3), we believe that the decoder of a VAE does the job it is designed to do. They cannot reconstruct images accurately, because the latent space of a VAE loses information about the image, by design. All we can do is imagine the type of image that a point in the latent space represents.

In the next section, we show examples of images generated by the LSR-GAN for both normal VAEs and $\beta$-VAEs (we also spend time describing VAEs, $\beta$-VAEs and the LSR-GAN in more detail). In addition, in this section we present a number of systematic experiments showing the performance of a VAE and LSR-GAN. In Section 3, we revisit the minimum description length formalism to explain why we believe a VAE is doomed to fail as a generative model. We conclude in Section 4. We cover more technical aspects in the appendices. In Appendix A we show that the correct loss function for a VAE requires minimising a term proportional to the logarithm of the mean squared error. In Appendix B we draw out the similarities and differences between our approach to hybridising VAEs with GANs and other work in this area. We present some additional experimental results in Appendix C. A detailed description of the architecture of LSR-GAN is given in Appendix D. We end the paper with Appendix E by showing some samples generated by randomly drawing latent variables and feeding them to the LSR-GAN.

## 2    IMAGINING LATENT SPACES

A natural question to ask is what information about an image gets represented in the latent space of a VAE. To answer this we can use the VAE encoder to generate a distribution $q_\phi(z|x)$ representing that image in the latent space (see Sections 2.1 for details on VAEs). From this distribution we can sample points in the latent space and feed this to the LSR-GAN generator. We show examples of this for both CIFAR-10 and ImageNet (down-sampled to $64 \times 64$) in Figure 1. In all cases in this paper the input images are taken from a test set that is independent of the training set. Note that both CIFAR-10 and ImageNet are "hard" for VAEs in the sense that they represent extremely diverse sets of images. As a consequence, the VAE latent space will struggle to store detailed information about the images and the VAE reconstructions will be poor. We have repeated this for a $\beta$-VAE (see section 2.3 for a full description of $\beta$-VAEs). We note that there is very little variation between the different samples drawn from $q_\phi(z|x)$, particularly for the standard VAE ($\beta = 1$), showing that the latent space of the VAE is relatively smooth (there is more variation when $\beta = 20$).

Input          Generated images based on Latent representation

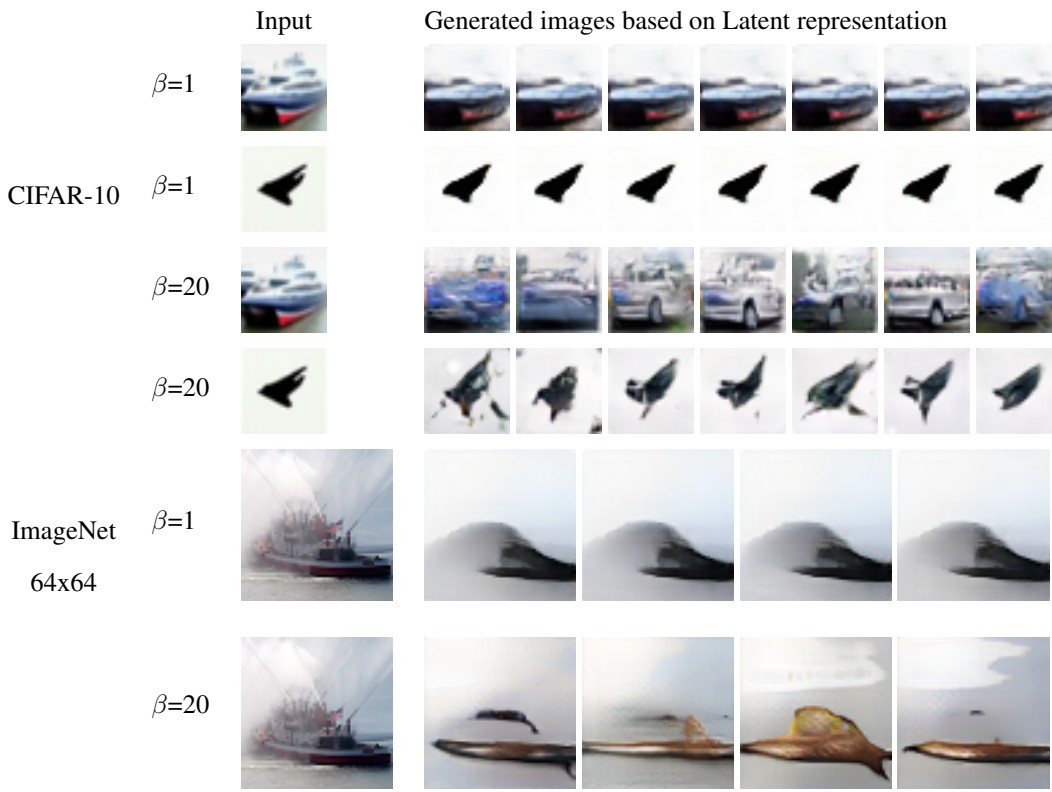

Figure 1: Images generated by our GAN from latent representations of an input image (test images) based on different $\beta$ value on CIFAR-10 and ImageNet, the first column is the original image.

To get a sense of the variation in the information stored in latent spaces we show in Figure 2 input-output pairs, where the left image is the input and right image is the output generated by the LSR-GAN generator seeded with a latent vector encoding of the input image. The reconstructions capture the shape and background, but clearly loses a lot of detail. In some cases it appears that the type of object is being captured, although in the case of the boat with the $\beta$-VAE (with $\beta = 20$) the wrong object is being rendered.

## 2.1 VARIATIONAL AUTOENCODERS

The structure of a VAE is represented schematically below.

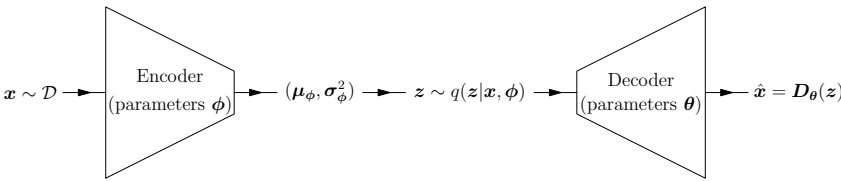

We sample an input $x$ from some dataset, $\mathcal{D}$. To be concrete we will consider the case where the inputs are images, although clearly a VAE can be used to represent many different types of data. For each input $x$ the encoder outputs a mean vector, $\mu$, and standard deviation vector, $\sigma$, that describes an axis aligned normal distribution, $q_\phi(z|x) = \mathcal{N}\Big(z\big|\mu_\phi(x), \mathrm{diag}(\sigma^2_\phi(x))\Big)$. A latent variable $z$ is sampled from this distribution and then fed to a decoder. For simple black and white datasets such as MNIST the decoder outputs a scalar at each pixel location that can be interpreted as the probability that the pixel is black. For more complex datasets the decoder ususal generates a "reconstruction" $\hat{x} = D_\theta(z)$. The probability of generating a pixel value $x_i$ is then usually taken as

CIFAR-10

ImageNet
64x64

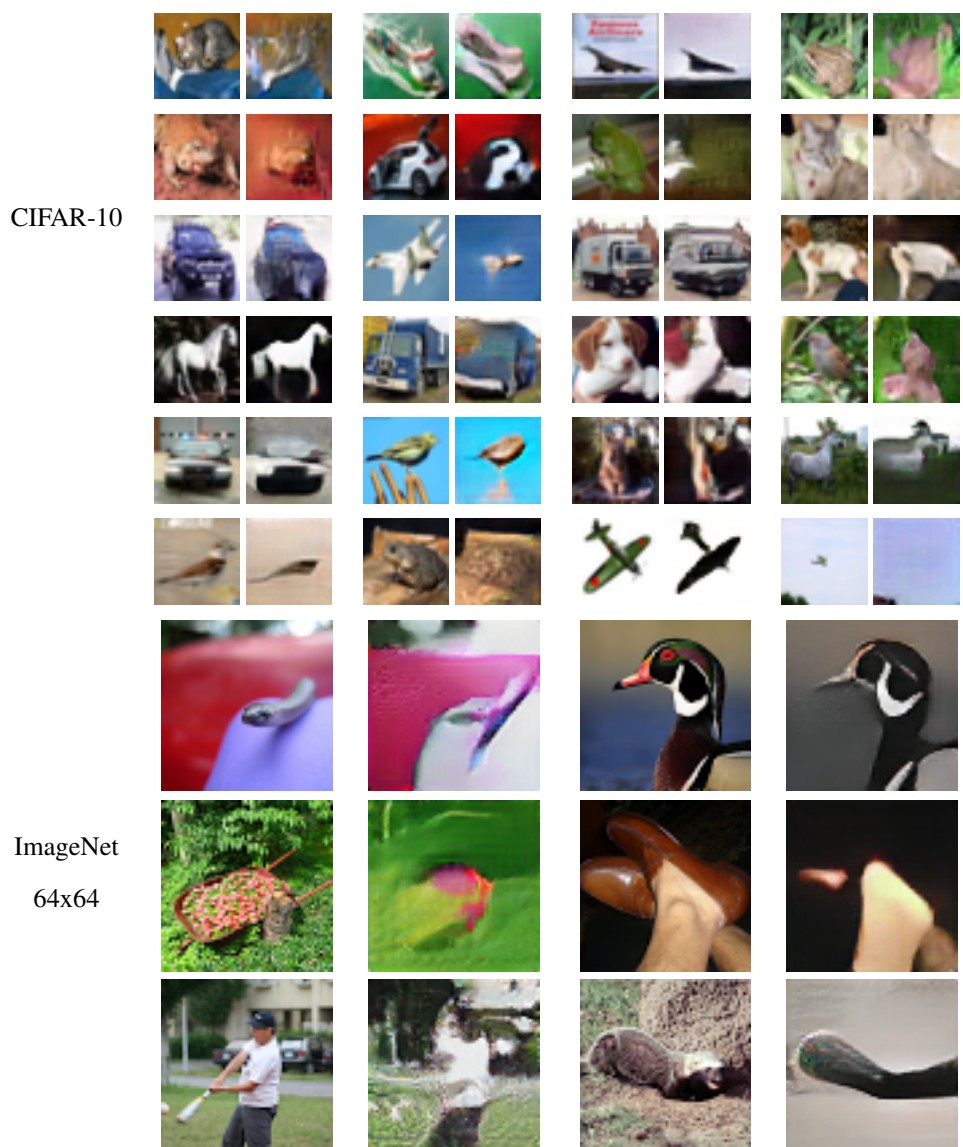

Figure 2: Examples of input-output pairs of images. The left image is an image from the test dataset. The right image is the image generated by the LSR-GAN seeded with a latent vector encoding of the input image ($\beta = 1$)

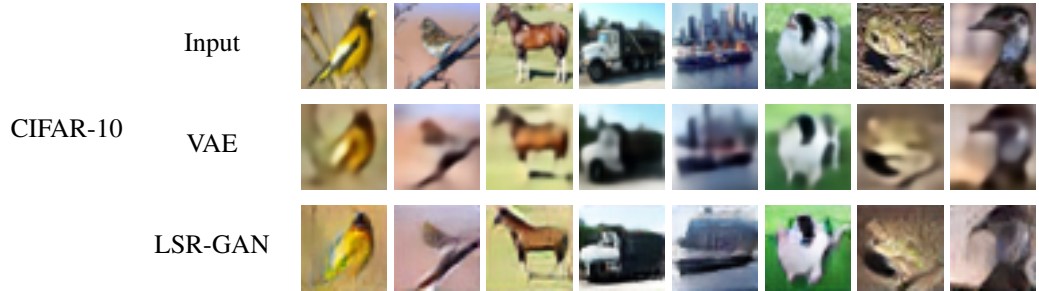

Figure 3: The comparison between reconstruction images of VAE and LSR-GAN on test dataset.

a normal distribution with mean $\hat{x}_i$ (i.e. $p_{\theta}(x_i|\boldsymbol{z}) = \mathcal{N}(x_i|\hat{x}_i, \sigma^2)$) and variance $\sigma^2$ that measures the expected size of the errors between the input images, $\boldsymbol{x}$, and the reconstructions, $\hat{\boldsymbol{x}}$.

The loss function for a VAE is equal to the negative evidence lower bound (ELBO)

$$\mathcal{L} = -\mathbb{E}_{\boldsymbol{x}\sim\mathcal{D}}\left[\mathbb{E}_{\boldsymbol{z}\sim q_{\phi}(\boldsymbol{z}|\boldsymbol{x})}[\log(p_{\theta}(\boldsymbol{x}|\boldsymbol{z}))] + \mathrm{KL}\left(q_{\phi}(\boldsymbol{z}|\boldsymbol{x})\|\mathcal{N}(\mathbf{0},\mathbf{I})\right)\right]. \tag{2}$$

As explained in Appendix A, $\log(p_{\theta}(\boldsymbol{x}|\boldsymbol{z}))$ is chosen to be proportional to the logarithm of the reconstruction error between $\hat{\boldsymbol{x}}$ and the input image $\boldsymbol{x}$—in our experiments this produced better reconstructions than replacing $\log(p_{\theta}(\boldsymbol{x}|\boldsymbol{z}))$ with the mean squared error.

## 2.2 LSR-GAN

LSR-GAN is a novel hybridization of VAE and GAN model. The most distinct difference of LSR-GAN from previous work is that it is a two-stage model. In the first stage we train the VAE model. Having done this we freeze the weights of the VAE and train the GAN. We train the discriminator, $D$, of LSR-GAN in the same way as a normal GAN. That is, we minimise a loss function

$$\mathcal{L}_{\mathrm{D}} = -\mathbb{E}_{\boldsymbol{x}}[\log(D(\boldsymbol{x}))] - \mathbb{E}_{\boldsymbol{z}}[\log(1 - D(G(\boldsymbol{z})))] \tag{3}$$

where $G$ is the generator or the decoder of LSR-GAN. The job of the discriminator, $D$ is, to decide whether its import is a real image or not. Thus, to optimise the loss function we neet to maximize the log-probability of passing the real data, $\boldsymbol{x}$, while minimising the log-probability of accepting a random sampling $G(\boldsymbol{z})$ generated by a generator $G$ seeded with a random latent vector $\boldsymbol{z}$. The architecture of the generator is the same as that of a normal GAN but the loss function is slightly different. We add an additional term giving

$$\mathcal{L}_{\mathrm{G}} = \mathbb{E}_{\boldsymbol{z}}[\log(D(G(\boldsymbol{z})))] + \lambda \log(q_{\phi}(\boldsymbol{z}|G(\boldsymbol{z}))). \tag{4}$$

The parameters of the discriminator and generator are trained in the usual tick-tock fashion using gradient descent. We built the VAE and the generator of GAN using a ResNet (He et al., 2016) as it gave slightly better performance than using a standard CNN. The architecture of the discriminator is the same as DCGAN (Radford et al., 2016). The architecture is described in Appendix D.

To test the LSR-GAN we use the VAE to generate a latent representation $\boldsymbol{z}$ for an image drawn from an independent test set. The latent vector is then used as a seed value for the generator in the LSR-GAN. The LSR-GAN can get sharper reconstruction images than VAE (see Figure 3). Although not visually so obvious, we have used a quantitative measure of sharpness computed as luminance-normalised Laplacian (San Pedro and Siersdorfer, 2009, Section 3.1.2). For the reconstructed images from the VAE we obtained a measure of $0.17 \pm 0.03$ while for the LSR-GAN we obtain $0.28 \pm 0.08$ (i.e. an improvement of a factor of two). We have also computed the FID measure of image quality for CIFAR-10 (Heusel et al., 2017). For images seeded from a testing example the VAE achieved a score of 89.8 while LSR-GAN achieved a score of 44.1, while for images seeded with random latent variable (i.e. $\boldsymbol{z} \sim \mathcal{N}(\mathbf{0}, \boldsymbol{I})$) the FID score for the VAE is 138.6 while for the LSR-GAN it is 47.4. This should not be surprising. The decoder of the VAE is training only where there are training images. Despite the fact that the KL-divergence tries to ensure that as much latent space as possible is used, the constraint of minimising the reconstruction loss means that most of the latent space is far from a training example. Although the VAE does not do too badly generating testing examples,

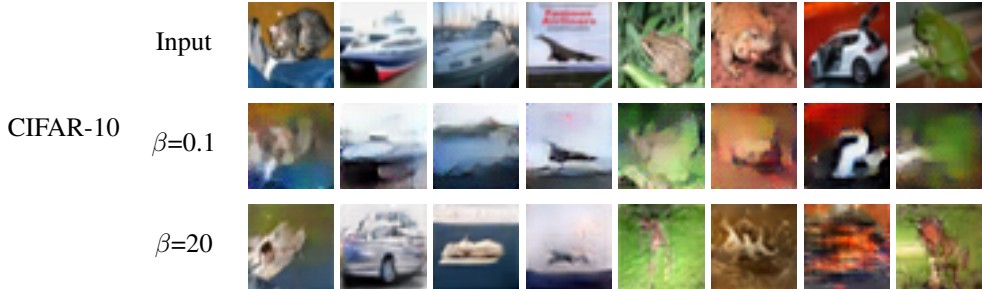

Figure 4: Examples of input-output pairs of images for different $\beta$ values for a $\beta$-VAE. The first row is from the test dataset. The second and third rows are Others reconstruction generated by LSR-GAN for VAEs trained with different $\beta$ values.

these tend to be substantially closer in the latent space to the training examples than random samples. In contrast the LSR-GAN is trained on random samples so that the generator will have to produce "realistic" images over the whole latent space. Of course, whether these generated images represents anything recognisable is open to question. For diverse training sets such as CIFAR-19 and ImageNet this may be very difficult. What image should we expect from a latent vector halfway between a truck and a bird? In Appendix E we show images generated by seeding LSR-GAN with random latent variables for CIFAR-10, ImageNet, MNIST and Celeb-A.

## 2.3 BETA-VAE

A goal of generating a latent representation is for the representation to be disentangled. Intuitively disentanglement seems clear: We would want information that is somehow causally independent to be encoded into orthogonal directions (or different variables) in our latent space (Bengio et al., 2013). Unfortunately, this is not only quite difficult to achieve in practice (at least, in an unsupervised setting), but it is even difficult to formulate (see Locatello et al. (2018)). Despite this difficulty, there have been many attempts to achieve disentanglement (Kim and Mnih, 2018; Chen et al., 2018; Burgess et al., 2018). One of the most prominent has been the $\beta$-VAE introduced by Higgins et al. (2017), where the KL-divergence term in a normal VAE is weighted by a parameter $\beta$

$$\mathcal{L} = -\mathbb{E}_{\boldsymbol{x} \sim \mathcal{D}} \left[ \mathbb{E}_{\boldsymbol{z} \sim q_{\boldsymbol{\phi}}(\boldsymbol{z}|\boldsymbol{x})} [\log \left( p_{\boldsymbol{\theta}}(\boldsymbol{x}|\boldsymbol{z}) \right)] + \beta \, \mathrm{KL} \left( q_{\boldsymbol{\phi}}(\boldsymbol{z}|\boldsymbol{x}) \| \mathcal{N}(\boldsymbol{0}, \mathbf{I}) \right) \right] . \tag{5}$$

The argument is that by making $\beta \gg 1$ we encourage disentanglement. Contrariwise, by making $\beta \ll 1$ we make a VAE closer to an auto-encoder. This improves the reconstruction performance on the training examples, but at the cost of allowing the latent space to over-fit the training set.

In Figure 4 we show examples of input-output pairs for different values of $\beta$. We observe that for large $\beta$ the outputs are quite different from the input images in contrast to small $\beta$ where many more details of the original input are captured.

Although the LSR-GAN model generates slightly clearer, less blurry, images, it has a lower reconstruction error than the VAE decoder. We show the mean squared error measured on a testing set from CIFAR-10 as a function of $\beta$ in Figure 5(a). This poor performance of the LSR-GAN is unsurprising, it uses the same information as the VAE (i.e. the information stored in the latent space). By producing sharper images it will pay the price of getting the boundary wrong. The blurry edges from the VAE is a way to hedge its bet and reduced the mean squared error. Interestingly, the mean squared error remains fairly constant as we increase $\beta$ from a low value, until we reach $\beta = 1$ after which it rapidly increases. One interpretation of this fact is that the VAE with $\beta = 1$ is successfully encoding all the useful information (i.e. compressible information) so for reconstructing unseen images it will perform as well as an auto-encoder. As we increase $\beta$ above 1, the reconstruction error increases rapidly.

In Figure 5(b) we show the classification performance as measured by a simple classifier trained on the CIFAR-10 training set. The classifier performance achieved an 84% correct classification on the raw images. We find little variation as we decrease $\beta$ below 1. As we increase $\beta$ above 1 the classification accuracy falls off. Again we can attribute this to the latent space of the VAE (with $\beta = 1$) capturing most useful information. Interestingly the high-$\beta$ VAE fails to capture

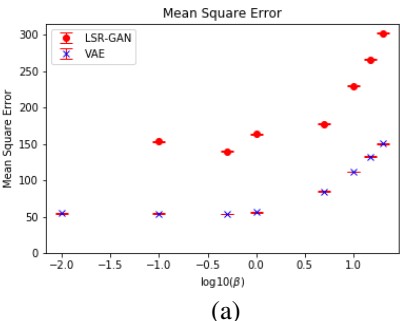 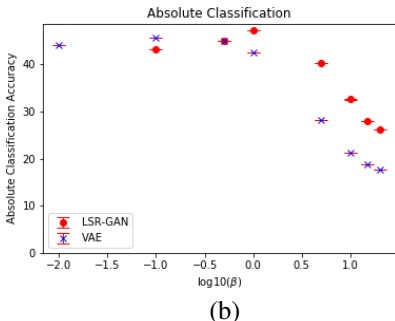

(a) (b)

Figure 5: Performance of the VAE (blue points) and LSR-GAN (red points) versus $\beta$. In (a) we show the means squared error, while in (b) we show the classification performance using a classifier taking the reconstructed images. The images are taken from CIFAR-10. Error bars show $\pm 1$ standard error.

"objectness" well. This suggests that, at least for CIFAR-10, the type of object does not contain very much information about its appearance and is rapidly discarded.

## 3 MINIMUM DESCRIPTION LENGTH

To understand what VAEs do it is useful to interpret them in the framework of the minimum description length (MDL) formalism. In MDL we consider communication a dataset $\mathcal{D}$ through a communication channel using as few bits as possible. We can do this using lossy compression, where we encode each input $\boldsymbol{x}$ by a code $\boldsymbol{z}$, which we communicate down our channel. The receiver decodes the message and produces an approximation of the input $\hat{\boldsymbol{x}}$. To communicate the original information we send the code $\boldsymbol{z}$ together with the error $\boldsymbol{\epsilon} = \boldsymbol{x} - \hat{\boldsymbol{x}}$ between the input $\boldsymbol{x}$ and the reconstruction $\hat{\boldsymbol{x}}$. Because the distribution of errors, $p(\boldsymbol{\epsilon})$, is more sharply concentrated than the original inputs, $p(\boldsymbol{x})$, this method allows us to communicate the image more efficiently than transmitting the raw pixel values. The expected cost of transmitting an input is

$$\mathcal{L} = \mathbb{E}_{\boldsymbol{x} \sim \mathcal{D}}[M(\boldsymbol{z}) + E(\boldsymbol{\epsilon})]$$

where $M(\boldsymbol{z})$ is the number of bits needed to communicate the code, $\boldsymbol{z}$, and $E(\boldsymbol{\epsilon})$ is the number of bits required to communicate the error, $\boldsymbol{\epsilon}$. In the MDL formalism we attempt to find a code that minimises the description length $\mathcal{L}$. To communicate the model and errors we need to use an optimal coding strategy. Rather than specifier and actual code we can use the Shannon bound (i.e. the negative log-probability of the tokens we transmit). For this to be meaningful, we need to specify both the errors and code to a finite precision. The precision of the errors will determine the accuracy of the data we communicate. If the $i^{th}$ component of the error is distributed according to $p(\epsilon_i)$ then the cost of communicating the error to a precision of $\Delta$ is approximately $-\log(p(\epsilon_i) \Delta) = -\log(p(\epsilon_i)) - \log(\Delta)$. The factor $-\log(\Delta)$ is common to all coding schemes so is irrelevant to choosing optimal codes $\boldsymbol{z}$. In contrast the precision to which we transmit the model will directly determine the cost $M(\boldsymbol{z})$. There is a balance to be struck: a more precise model can potential lead to a better reconstruction $\hat{\boldsymbol{x}}$, reducing the reconstruction cost, $E(\boldsymbol{\epsilon})$, but at the same time increasing the cost, $M(\boldsymbol{z})$, of communicating the code $\boldsymbol{z}$.

The KL-divergence term, $\mathrm{KL}\left(q(\boldsymbol{z})\middle\|p(\boldsymbol{z})\right)$ (also known as the relative entropy) can be interrupted as the communication cost (in nats) of transmitting a random variable $\boldsymbol{z}$ with uncertainty given by $q(\boldsymbol{z})$ assuming an underlying probability distribution of all random variables of $p(\boldsymbol{z})$. Using this interpretation we see that the loss function of a VAE is equivalent to the expected message length (in nats) of communicating a sample from the dataset $\mathcal{D}$ by using a random variable $\boldsymbol{z}$ with uncertainty $q(\boldsymbol{z})$. By minimising the loss function we find a coding scheme with the minimum description length (or, at least, an approximate local minimum). By encoding a message as a random variable $\boldsymbol{z}$ drawn from a distribution $q_{\boldsymbol{\phi}}(\boldsymbol{z}|\boldsymbol{x})$ the VAE is able to find an optimal balance between accuracy to which it transmits the model (determined by the standard deviation vector, $\boldsymbol{\sigma}$, generated by the VAE encoder) and the need to reduce the reconstruction error. *From an MDL perspective the ELBO is the correct objective function, and should not be regarded as a approximate lower bound to what we really want to achieve.* If there are too many dimensions in the latent space then some of the

components of $z$ (channel in information theory terms) are such that $z_i$ is approximated distributed by $\mathcal{N}(z_i|0, 1)$ for all inputs $x$. The channel is effectively "switched off" (and it will be ignored by the decoder as it is just a source of random noise). This is referred to as latent variable collapse and is sometimes viewed as problematic, however, from the MDL viewpoint it acts as an elegant automatic dimensionality selection technique.

The job of the decoder in a variational autoencoder is to reconstruct the image only using information that can be compressed. Image specific information is ignored. For example, information about the precise shape of an object is probably not compressible. As a result the decoder tends to hedge its bets and has a blurry outline. Of course, some encoders and decoders will be better than others, but to date there is little evidence in the literature that the performances of VAEs are massively sub-optimal, at least, when working with images. With an extremely powerful encoder and decoder and a limited dataset it would be possible for the encoder to communicate an identifier of the input image and for the decoder to reproduce the image just from the identifier, thus avoiding communicating any information about the visual content of the image—this requires that the decoder memorises all the images. This would be an extreme case of what is sometimes called posterior collapse. There is some evidence that with very strong encoders and decoders that the amount of information stored in the latent space (as measured by the KL-divergence) decreases (Bowman et al., 2015). This might point to a weakness of the VAE set-up—the MDL set-up really only makes sense when the dataset is arbitrarily large—, but this problem could be ameliorated by data augmentation. However, using standard CNN encoders and decoders we found no evidence for memorisation of the images (for example, the VAE would produce a similar level of reconstruction for images from a separate test set). For language modelling there seems to be more evidence that VAEs often fail to extract information in the latent space, but for images it seems likely that a properly trained VAE will extract a good fraction of the compressible information. We believe that the failure of the VAE decoder to produce high quality reconstructions (except in the case very of simple datasets such as MNIST and possibly CELEB-A) is because to do so would require communicating information that is non-compressible. As a consequence we should not think of the decoder of a VAE as a generative model: It will, by design, produce blurry and poor quality reconstructions. We want this to ensure that the latent space only captures information that is common across many images. We see the mapping from images to latent space as a many-to-one mapping. Thus, the mapping from the latent space to images will be ambiguous and the best we can do is imagine an image compatible with the latent variable: exactly what we have designed the LSR-GAN to do.

## 4 CONCLUSION

VAEs are often taken to be a pauper's GAN. That is, a method for generating samples that is easier to train than a GAN, but gives slightly worse results. If this is the only objective then it is clearly legitimate to modify the VAE in anyway that will improve its performance. However, we believe that this risks losing one of their most desirable properties, namely their ability to learn features of the whole dataset while avoiding encoding information specific to particular images. We have argued that because of this property, a VAE is not an ideal generative model. It will not be able to reconstruct data accurately and consequently will struggle even more with generating new samples. One of the weaknesses of the vast literature on VAEs is that it often attempts to improve them without regard to what makes VAEs special.

As we have argued in this paper, a consistent way of using the latent space of a VAE is to use a GAN as a data renderer, using the VAE encoder to ensure that the GAN is generating images that represent the information encoded in the VAE's latent space. This involves "imagining" the information that the VAE disregards. LSR-GAN can be particularly useful in generating random samples, although, as shown in Appendix E, for very diverse datasets the samples are often not recognisable as real world objects. Although there are already many VAE-GAN hybrids, to the best of our knowledge, they are all designed to "fix" the VAE. In our view VAEs are not broken and "fixing" them is actually likely to break them (i.e. by encoding image specific information in the latent space). Although, the main idea in this paper is relatively simple, we believe its main contribution is as a corrective to the swath of literature on VAEs that, in our view, often *throws the baby out with the bath water* in an attempt to fix VAEs despite the fact that perform in exactly the way they were designed to.

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

# Supplementary Material

## A  ON THE ELBO

In the standard VAE we maximise the log-probability of generating the original image. In the original paper this was achieved by the decoder outputting a probability distribution akin to what happens in the latent space. More often it is assumed that the pixel errors are normally distributed with some variance $\sigma^2$. Thus the log-probability of generating all the images is

$$\sum_{\boldsymbol{x}\sin\mathcal{D}}\mathbb{E}_{\boldsymbol{z}\sim q(\boldsymbol{z}|\boldsymbol{x})}[\log\left(p(\boldsymbol{x}|\boldsymbol{z})\right]=\sum_{i=1}^{N}\log\left(\mathcal{N}(\boldsymbol{x}|\hat{\boldsymbol{x}},\sigma^2)\right)=-\sum_{i=1}^{N}\frac{(x_i-\hat{x}_i)^2}{2\sigma^2}-\frac{N}{2}\log\left(2\pi\sigma^2\right)$$

where the sum is over all predicted pixels—i.e. the number of pixels in an image times the number of colour channels times the number of examples (or, more usually, the mini-batch size). However,

$$\sigma^2=\frac{1}{N}\sum_{i=1}^{N}(x_i-\hat{x}_i)^2$$

(at least, if we make the natural assumption that the errors have mean zero). As a consequence

$$\sum_{\boldsymbol{x}\in\mathcal{D}}\mathbb{E}_{\boldsymbol{z}\sim q(\boldsymbol{z}|\boldsymbol{x})}[\log\left(p(\boldsymbol{x}|\boldsymbol{z})\right]=-\frac{N}{2}-\frac{N}{2}\log\left(2\pi\sigma^2\right)$$

so that we should minimise $N\log\left(\sigma^2\right)/2$. In information theory terms this tells us that it cheaper to communicate the residues if they are more tightly concentrated. Note that since $\sigma^2$ is proportional to the mean squared error, $E_{\mathrm{MSE}}$, it suffices to minimise $N\log\left(E_{\mathrm{MSE}}\right)/2$. We note that

$$\frac{\partial}{\partial\hat{x}_i}\frac{N}{2}\log\left(2\pi\sigma^2\right)=\frac{\hat{x}_i-x_i}{\sigma^2}$$

which is precisely the gradient of

$$\sum_{i=1}^{N}\frac{(x_i-\hat{x}_i)^2}{2\sigma^2}$$

if we ignored the dependence of $\sigma^2$ on $\hat{x}_i$. In many publically available implementations of VAEs the algorithm minimises $\sum_{i=1}^{N}(x_i-\hat{x}_i)^2$ which arbitrarily assumes $\sigma^2=\frac{1}{2}$ rather than its true value. This means that these implementations are effectively running a $\beta$-VAE with some unknown $\beta$ (in our experience with $\beta>1$). This makes comparing results from different VAE implementations difficult. For example, rescaling outputs to lie in the range $[-1,1]$ rather than $[0,1]$ would change the effective $\beta$-value.

## B  VAE-GAN HYBRIDS

The hybridisation of VAE (or autoencoder) and GAN models have been developed for several years. There are many attempts on this area and we compare LSR-GAN to the most related work in this section.

The adversarial types autoencoder is the most intuitive and simplest way to combine a VAE or an autoencdoer and a GAN models. Most of these models introduce a discriminator into the autoencoder training. AAE (Makhzani et al., 2016) applies a discriminator to distinguish the output of encoder and the random sample from the prior distribution. It uses this discriminator to replace the KL term in VAE. VAE/GAN (Larsen et al., 2016) is the first model that applied feature-wise errors and the input of its generator contains three different types images: the reconstruction images, the generated images and the real images. The same as our model, it collapse the decoder and the generator into one. MDGAN (Che et al., 2017) is another AE-GAN hybrid which is close to VAE/GAN, they try to match the manifold of GAN to real data by adding a geometric metrics regulariser and mode regulariser. None of these methods feed the output of generator back into the encoder or train their

network in two-stages, which is the biggest difference between these methods and ours. Also, many of these hybrid models adopt an autoencoder instead of VAE while the VAE in our model cannot be replaced by an autoencoder.

There are not many models that use the output of decoder to feed the encoder. The Introspective Adversarial Network (IAN) (Brock et al., 2017) is a unified model which means the discriminator is not separate. IAN only encodes the feature that extracted by discriminator rather than the raw images. The discriminator of IAN extracts features from both raw images and synthetic images. The generator accept both random sample and the output of the discriminator as inputs at the same time. In contrast, our models only accept one input. Another model that adopts the introspective method is IntroVAE (Huang et al., 2018), it constructs the inference model $E$ and generator model $G$ in a circulation loop. IntroVAE has the ability to generate high-resolution images. But it does not contain any discriminator network.

The most closely work to our LSR-GAN is VEEGAN (Srivastava et al., 2017). It introduces a second network $F_\theta$ to the GAN. The task of $F_\theta$ is to map both the real images and synthetic images to a Gaussian distribution which is what we ask the encoder to do. When the input of $F_\theta$ is the output of generator, the objective function minimise the distance between the input of generator and the output of $F_\theta$. If the input of $F_\theta$ is real data, the objective function minimise the cross entropy between Gaussian prior and the output of $F_\theta$. Another related model is the Generative moment matching networks (GMMN) (Li et al., 2015). In this model the autoencoder is frozen and they then minimize the maximum mean discrepancy (MMD) between the generated representation and data representation, and they use an uniform prior to generate the representations. In LSR-GAN, we match two Gaussian distributions in maximizing the probability distance. None of these related works are two-stages models except GMMN. Also, to the best of our knowledge, LSR-GAN is the first VAE-GAN hybrid model that applies the probability distance in the loss function.

## C ADDITIONAL EXPERIMENTS

We briefly present some additional experimental data.

### C.1 DEPENDENCE OF LSR-GAN ON $\beta$

In Table 1 we present measurements of the performance of outputs from both VAEs and LSR-GAN for different values of $\beta$. Some of this data is also presented graphically in Figure 5, but we have included additional measurements.

Table 1: The measurement for different $\beta$ values. Variance is the variance among images generated by same latent representations. Absolute accuracy is the rate that classifier classifies reconstruction images right. Relative accuracy is the rate that classifier classifies reconstruction images the same as raw images.

| | $\beta$=0.01 | $\beta$=0.1 | $\beta$=0.5 | $\beta$=1 | $\beta$=5 | $\beta$=10 | $\beta$=15 | $\beta$=20 |
|---|---|---|---|---|---|---|---|---|
| Mean Square Error (VAE) | $54.56 \pm 0.30$ | $54.07 \pm 0.29$ | $53.80 \pm 0.29$ | $55.9 \pm 0.3$ | $84.64 \pm 0.42$ | $111.49 \pm 0.53$ | $132.56 \pm 0.61$ | $150.40 \pm 0.66$ |
| Mean Square Error (LSR-GAN) | Model Collapse | $153.09 \pm 0.66$ | $139.75 \pm 0.63$ | $163.06 \pm 0.73$ | $177.22 \pm 0.86$ | $229.53 \pm 1.07$ | $265.74 \pm 1.19$ | $302.17 \pm 1.47$ |
| Variance (VAE) | $3.03e-5 \pm 2.44e-7$ | $2.7e-3 \pm 1.89e-5$ | $0.05 \pm 3e-4$ | $0.17 \pm 0.01$ | $4.94 \pm 0.03$ | $21.02 \pm 0.13$ | $50.80 \pm 0.34$ | $89.25 \pm 0.61$ |
| Variance (LSR-GAN) | Model Collapse | $0.04 \pm 4.3e-3$ | $0.23 \pm 2.8e-3$ | $1.67 \pm 0.02$ | $416.83 \pm 5.10$ | $2667.45 \pm 37.73$ | $4920.23 \pm 47.43$ | $4234.5 \pm 58.73$ |
| Absolute Classification (VAE) | $44.08 \pm 0.22\%$ | $45.66 \pm 0.21\%$ | $45.01 \pm 0.17\%$ | $42.56 \pm 0.13\%$ | $28.17 \pm 0.13\%$ | $21.26 \pm 0.20\%$ | $18.89 \pm 0.18\%$ | $17.70 \pm 0.23\%$ |
| Absolute Classification (LSR-GAN) | Model Collapse | $43.29 \pm 0.18\%$ | $45.06 \pm 0.23\%$ | $47.19 \pm 0.13\%$ | $40.23 \pm 0.18\%$ | $32.58 \pm 0.21\%$ | $27.93 \pm 0.21\%$ | $26.18 \pm 0.17\%$ |
| Relative Classification (VAE) | $45.69 \pm 0.21\%$ | $44.11 \pm 0.20\%$ | $43.38 \pm 0.21\%$ | $43.75 \pm 0.25\%\%$ | $28.34 \pm 0.15\%$ | $21.59 \pm 0.21\%$ | $18.90 \pm 0.18\%$ | $16.90 \pm 0.21\%$ |
| Relative Classification (LSR-GAN) | Model Collapse | $45.02 \pm 0.18\%$ | $44.71 \pm 0.16\%$ | $48.41 \pm 0.25\%\%$ | $43.54 \pm 0.18\%$ | $37.90 \pm 0.17\%$ | $33.68 \pm 0.20\%$ | $31.91 \pm 0.19\%$ |

### C.2 DEPENDENCE OF LSR-GAN ON $\lambda$

The performance of the LSR-GAN depends on the hyper-parameter $\lambda$. This balances the need to produce convincing images (from the discriminator's point of view) with the requirement that the latent space of the GAN should be close to that for the VAE. These two objectives are not necessarily contradictory, although we will see that changing $\lambda$ has benefits and drawbacks.

In Figure 6 we show the effect of changing $\lambda$ over approximately three orders of magnitude on (a) the absolute classification accuracy (b) the classification accuracy compared to the class labels

Table 2: The measurement for different $\lambda$ values. Variance is the variance among the images generated by same latent representations. Absolute accuracy is the rate that classifier classifies reconstruction images right. Relative accuracy is the rate that classifier classifies reconstruction images the same as raw images.

| | $\lambda$=0.01 | $\lambda$=0.1 | $\lambda$=0.5 | $\lambda$=1 | $\lambda$=5 | $\lambda$=10 | $\lambda$=15 | $\lambda$=20 |
|---|---|---|---|---|---|---|---|---|
| Mean Square Error | $285.99 \pm 1.39$ | $231.34 \pm 1.14$ | $193.72 \pm 0.90$ | $163.06 \pm 0.73$ | $104.15 \pm 0.46$ | $90.66 \pm 0.43$ | $86.81 \pm 0.42$ | $84.69 \pm 0.40$ |
| Variance | $68.54 \pm 1.17$ | $15.06 \pm 0.25$ | $3.96 \pm 0.08$ | $1.67 \pm 0.02$ | $0.70 \pm 0.01$ | $0.54 \pm 0.00$ | $0.50 \pm 0.00$ | $0.48 \pm 0.00$ |
| Absolute Classification | $35.52 \pm 0.23\%$ | $45.2 \pm 0.21\%$ | $46.96 \pm 0.28\%$ | $47.19 \pm 0.13\%$ | $47.21 \pm 0.20\%$ | $47.45 \pm 0.18\%$ | $47.89 \pm 0.21\%$ | $48.72 \pm 0.23\%$ |
| Relative Classification | $36.37 \pm 0.18\%$ | $46.40 \pm 0.20\%$ | $48.08 \pm 0.21\%$ | $48.41 \pm 0.25\%\%$ | $48.65 \pm 0.15\%$ | $48.72 \pm 0.30\%$ | $48.79 \pm 0.28\%$ | $50.22 \pm 0.21\%$ |

predicted by the classifier on the raw images (c) the mean squared reconstruction error and (d) the variance in the predictions when choosing different samples from $q_\phi(z|x)$. We see that increasing $\lambda$ improves the classification performance (both relative and absolute). However, and perhaps surprisingly, increasing $\lambda$ produces a significant reduction in the reconstruction error. More intuitively it also causes a reduction in the variance between images sampled independently from $q_\phi(z|x)$. That is, using the encoder in the LSR-GAN acts a regulariser ensuring close by points in latent space map to similar images. More details are given in Table 2.

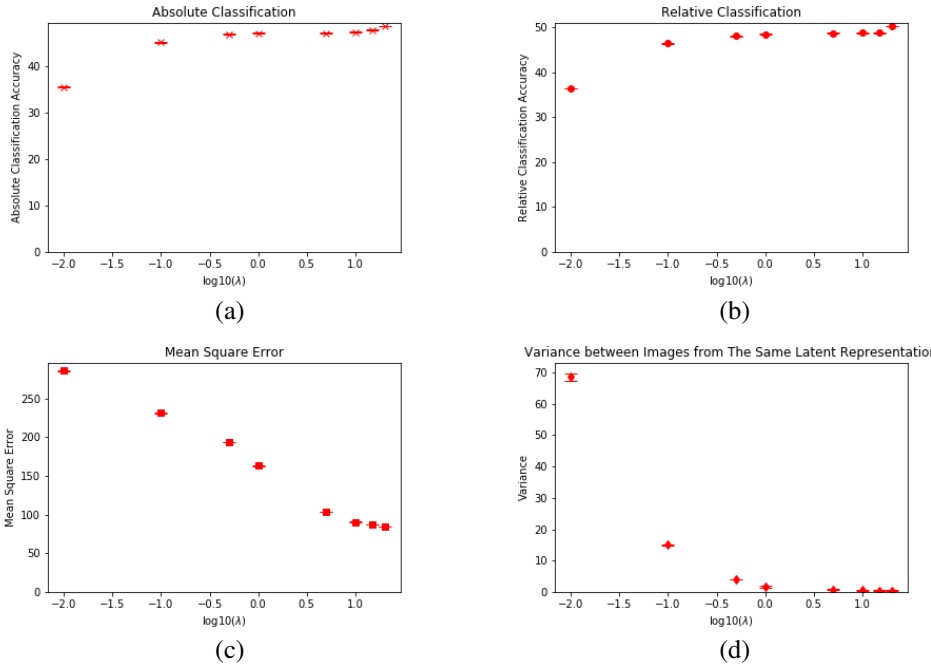

Figure 6: Graphs showing the classification performance of images generated by our GAN with different $\lambda$ values (on test dataset). The left-bottom graph shows the mean square error between reconstruction images and imagination images. The right-bottom graph shows the variance between images generated by our GAN from latent representations of an input image. The x-axis is the log value of different $\lambda$, the errors are too small which make error bars look like lines.

## D   ARCHITECTURE OF THE LSR-GAN

In this appendix we describe the detailed architecture of the VAE and LSR-GAN we used. Table 3 describes the structure of the VAE's encoder and decoder and the GAN's generator and discriminator networks. The encoder and decoder/generator are based on a ResNet. The ResNet block structure is shown in Figure 7. Both networks are optimized using Adam (Kingma and Ba, 2015) with a learning rate of $2 \times 10^{-4}$ and $\beta_1 = 0.5$. The code we used to implement the models is available at https://github.com/iclr-2020-zzz/LSR-GAN.

Table 3: Architectures for the three networks in LSR-GAN, BN denotes batch normalization

| Encoder | Decoder(Generator) | Discriminator |
|---|---|---|
| ResBlock down 64 | 512 FC Network, BN, ReLU | 4 x 4, stride=2, padding=1, 32 CNN, LeakyReLU |
| ResBlock down 128 | ResBlock up 128 | 4 x 4, stride=2, padding=1, 64 CNN, BN, LeakyReLU |
| ResBlock down 32 | ResBlock up 64 | 4 x 4, stride=2, padding=1, 128 CNN, BN, LeakyReLU |
| 128 FC Network, BN, ReLU | ResBlock up 32 | 4 x 4, stride=2, padding=1, 3 CNN, Sigmoid |
| 128 FC Network | 1 x 1, 3 CNN, Tanh | |

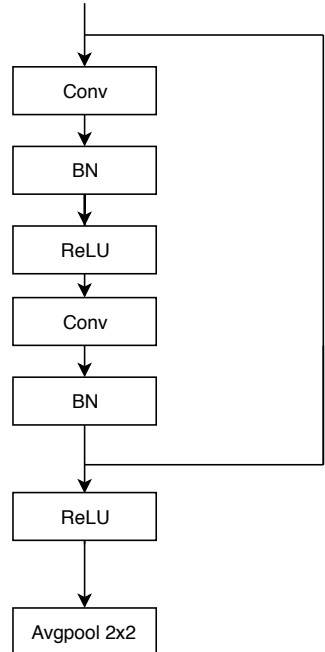

Figure 7: ResBlock architecture.

# E  SAMPLING

In this appendix we show sample images generated by LSR-GAN starting with a random seed $z \sim \mathcal{N}(\mathbf{0}, \mathbf{I})$. These are shown in Figure 9 for an LSR-GAN trained on CIFAR-10 and ImageNet. Although the images superficially look reasonable on close inspection it is clear that most samples for the LSR-GAN trained on CIFAR-10 and ImageNet are not real world objects. This reflects the fact that the images for these two dataset are very variable leaving most of the latent space representing rather surreal objects.

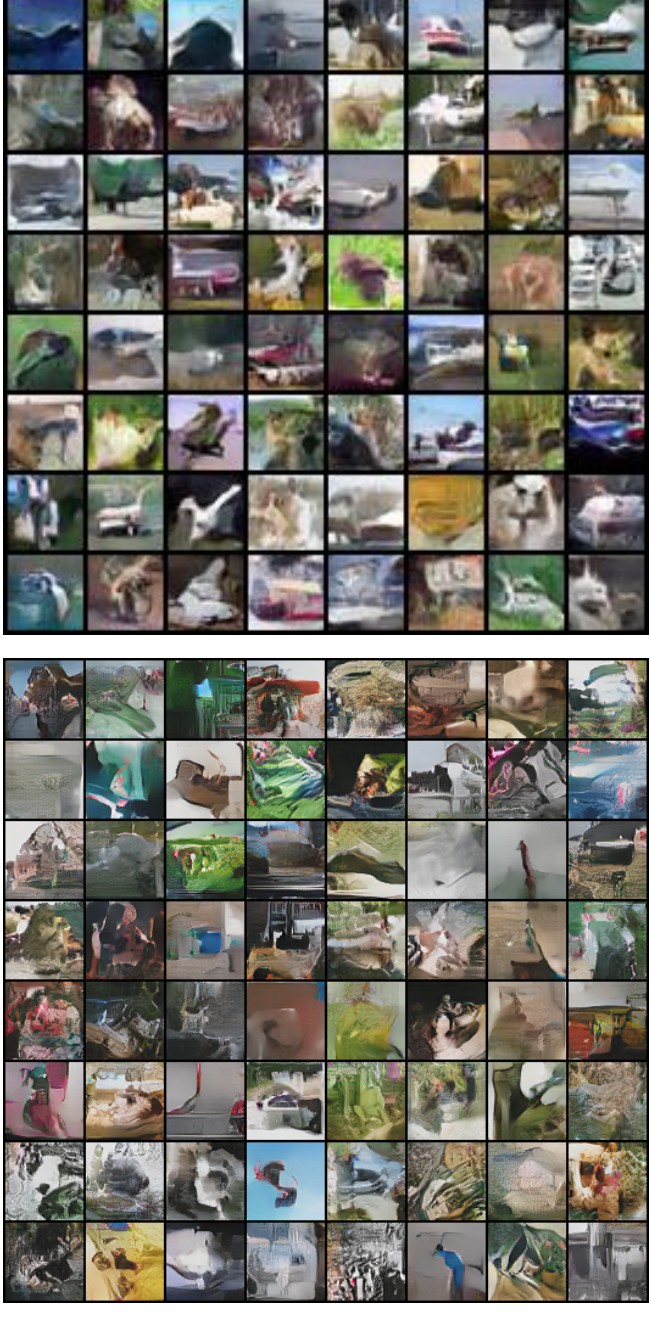

Figure 8: Random samples generated by LSR-GAN trained on CIFAR10 and ImageNet with $\beta$ =1 and $\lambda$=1.

We have also trained LSR-GAN on MNIST and Celeb-A with samples shown in Figure 9. Perhaps unsurprisingly, most samples are identifiable.

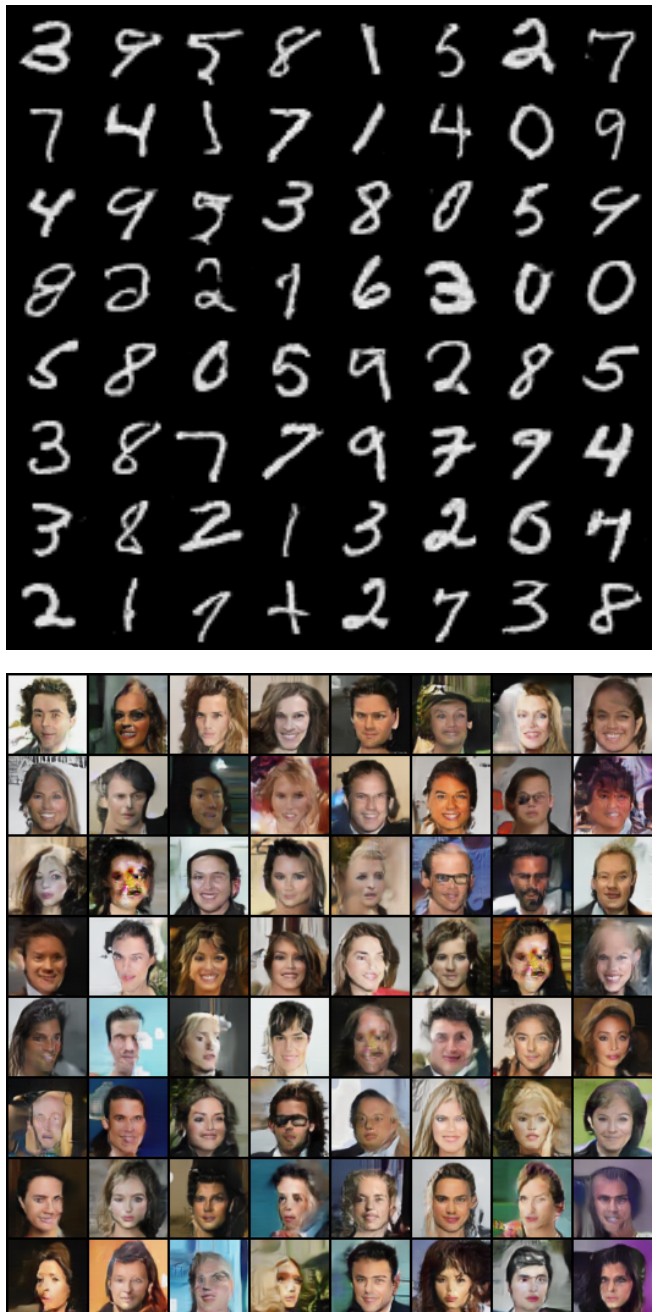

Figure 9: Random samples generated by LSR-GAN trained on MNIST and Celeb-A with $\beta$ =1 and $\lambda$=1.

