# OpenReview forum: "Imagining the Latent Space of a Variational Auto-Encoders"
_ICLR.cc/2020/Conference — Reject_

### Official Review · AnonReviewer3 · 2019-10-07
**Official Blind Review #3**

**Rating:** 3

**Review:**

This paper proposes to augment the VAE objective with two additional terms: (1) a GAN-like objective that ensures that the generated samples are not distinguishable from real samples, and (2) an additional regularizer to make sure that the latent variable that generated the image can be reconstructed from the VAE encoder.

Overall the paper is written very well and was a pleasure to read. In particular I appreciated the differences between the present work and the many VAE/GAN hybrids in section B of the Appendix. However, after reading the paper a couple of times, it was still not clear to me what the "main point" of the paper is. To be more specific:

- From the perspective of obtaining good unconditional samples from a VAE-like model, the authors do not compare their approach against like methods like IntroVAE (https://arxiv.org/pdf/1807.06358.pdf) which also adds an adversarial objective to the VAE. Qualitatively at least, it seems apparent that the generated samples here are worse in quality that the IntroVAE work.

- From the perspective of learning more faithful reconstructions, it seems clear from Figure 5(a) that a regular beta-VAE does better. And to me it is not clear why we would want faithful reconstructions?

- The main novelty of the proposed method is in adding an extra term to the generator loss controlled by lambda (equations 1 and 4). In the appendix the authors experiment with varying the lambda parameter and find that generally having lambda > 0 produces "better" metrics, though the choice of these metrics are somewhat questionable. It would be great to see the generated samples as \lambda is varied (to me this is more interesting than seeing the generations as \beta is varied).

Further, I take several issues with the authors' point of view regarding the current state-of-affairs in VAEs:

1. "However, one of their perceived problems is their reconstruction performance."

I somewhat disagree with this characterization. Sure, there has been much work on modifying the VAE objective such that the latent variable is not ignored (i.e. posterior collapse), but the point of these works is not to get "better reconstruction performance".

2. "However, having a model that does not over-fit the dataset can be useful, but in this case the decoder of a standard
VAE should not be regarded as a generative model—that is not its purpose. If we wish to generate
realistic looking images we need to imagine the information discarded by the encoder"

I am not sure I understand this characterization. The decoder is by definition a generative model. Depending on the decoder/encoder capacities (e.g. PixelCNN decoder vs DeConvNet decoder), different types of information will be encoded in the latent space.

3. "The job of the decoder in a variational autoencoder is to reconstruct the image only using information
that can be compressed. Image specific information is ignored. For example, information about the
precise shape of an object is probably not compressible. As a result the decoder tends to hedge its
bets and has a blurry outline."

Again, all of this is dependent on how the encoder/decoder is parameterized. I do not agree that "information about the precise shape of an object is probably not compressible". For example see https://hal.archives-ouvertes.fr/hal-01676326/document

4. "VAEs are often taken to be a pauper’s GAN. That is, a method for generating samples that is easier
to train than a GAN, but gives slightly worse results."

My view on VAEs is that they are a way of training latent variable models with likelihood training. One potential application of this is to generate samples, but that is not the only (nor the primary) application.


Finally, I hope I am not coming across as nitpicking or overly combative, but I am genuinely confused as to the problem that this paper is addressing. I look forward to discussing further among other reviews and the authors during the rebuttal period.

[Response to author rebuttal]

Thank you very much for your thoughtful response. However, I must say that I just fundamentally disagree with motivations and some of the statements made in the paper. In particular, there seems to be some conflation (I could be misunderstanding) of "good reconstruction" vs "good generation". For example:

- "Our point is that it does not generate a realistic image by design (in contrast to a GAN which is designed to generate a realistic image)."

I am not sure I understand. Would you say an autoregressive model (e.g. PixelCNN) does not generate realistic images by design? (Clearly, they do!). Would you say that autoregressive language models (e.g. GPT2) do not generative realistic language by design? The argument seems to be that likelihood-based training of generative models does not, by design, encourage realistic-looking images (indeed it is true that good likelihood does not *necessarily* imply good generative models). However, there is massive empirical evidence that likelihood training does result in good generation.

After reviewing the other reviews and the author rebuttals, I am maintaining my original score.




**Experience Assessment:**

I have read many papers in this area.

**Review Assessment: Checking Correctness Of Derivations And Theory:**

I assessed the sensibility of the derivations and theory.

**Review Assessment: Checking Correctness Of Experiments:**

I assessed the sensibility of the experiments.

**Review Assessment: Thoroughness In Paper Reading:**

I read the paper at least twice and used my best judgement in assessing the paper.

---

> ### Author Response · Authors · 2019-11-08
> **Response to Reviewer #3**
>
> We are glad that the paper was a pleasure to read.  We feel it was taken
> in the spirit it was written as a thought provoking reflection on VAEs.
> We are sad that in the end the reviewer did not see the point.  Clearly,
> this was a failure of our paper, but we feel that one of the problems
> was that the main novelty was taken to be LSR-GAN, which for us was just
> a visualisation tool.  The main novelty was the interpretation of the
> VAE and drawing conclusions from that interpretation.
>
> To clarify why LSR-GAN is useful.  Although VAEs give reasonable
> reconstructions on training and test data, for many data sets they have
> poor performance in generating images from randomly sampled parts of the
> latent space.  Clearly this is because the VAE decoder has not been
> trained in these regions.  However, these are regions of great interest
> (the latent representation of VAEs are there real strength and we would
> like to these to be meaningful away from training images).  LSR-GAN give
> us much better images when we sample from the latent space (as evidenced
> by FID scores for example).  Due to page limits we had to relegate
> examples of these to Appendix E, but these are remarkably more realistic
> (to be fair surreal) than we would get from using the VAE decoder.
>
> *** 1.
>
> We don't quite see what you are disagreeing about.  We called this a
> "perceived problem" because, as we strongly argued, VAEs are not
> designed for reconstruction.  Clearly a lot of work on VAEs is focused
> on the latent representation they learn.  Nevertheless, some of the
> literature has focused on improving performance.  We have no desire to
> belittle work, for example, on preventing latent variable collapse, but
> we also think it is worth stating that this could also be viewed as
> automatic dimensionality selection mechanism, which is a desirable
> property rather than a problem that needs to be fixed.
>
> *** 2.
>
> We are trying to present a new perspective and in doing so we wanted to
> make a strong (provocative?) statement.  We accept that the decoder is a
> generative model in that it generates something.  Our point is that it
> does not generate a realistic image by design (in contrast to a GAN
> which is designed to generate a realistic image).  Undoubtedly different
> information will be stored in the latent space depending on the decoder,
> but "incompressible" information (e.g. information that appears only in
> one training example) won't be stored for any decoder.
>
> *** 3.
>
> We tried to give a concrete example (inserting the word probably because
> we are speculating).  Our point was to explain why you would expect some
> blurriness.  A precise description of an image would seem incompressible
> (otherwise VAEs should achieve zero reconstruction error which in our
> experience is quite far from what we find).
>
> *** 4.
>
> We totally agree!  We concede that it would be useful to make this
> explicit.  Our whole paper is to point out that a VAE shouldn't be
> viewed as a pauper's GAN.  By design the decoder is not meant to produce
> good reconstructions.
>
> We have clearly failed to communicate the purpose of this paper.  We
> believe the MDL interpretation of VAEs provide many novel insights.  We
> cannot claim the MDL interpretation is new, but the insights are not
> well known.  They include the observation that the decoded images should
> be poor quality, that latent variable collapse is desirable, that latent
> spaces need to imagined because they won't capture all the information
> in an image and even that the correct form of the reconstruction term is
> $N \log(\sigma) + const$ (i.e. the entropy of $N$ independent variables from a
> distribution $N(0,\sigma^2)$).  These are not revolutionary, but at the same
> time they are not well known and we feel they are important.

---

### Official Review · AnonReviewer2 · 2019-10-23
**Official Blind Review #2**

**Rating:** 1

**Review:**

Summary:
This paper proposes a hybrid VAE-GAN model, called the latent space renderer-GAN (LSR-GAN), with the goal to “imagine” the latent space of a VAE, and to improve the decoding and sampling quality of a VAE. First, a VAE-like model is trained, after which the encoder weights are frozen, and the decoder is trained as the generator of a GAN (together with an auxiliary discriminator). The generator loss also contains a reconstruction-like term in the latent space, described by the negative log density of the encoding distribution of the original latent conditioned on the output of the generator: -log q(z|g(z)).

Decision: reject
This paper contains incorrect claims. Leaving those mistakes aside, the experiments don’t lead to new insights and no comparison against other VAE-GAN hybrids is made.

Supporting arguments for decision:
In the introduction the authors state “This is a consequence of the well known evidence lower bound or ELBO objective function consisting of a negative log-probability of generating the original image from the latent representation (this is often implemented as a mean squared error between the image and the reconstruction, although as we argue in Appendix A this term should be proportional to the logarithm of the mean squared error)...”. This statement is surprising and I don’t see how it can be correct. If -log p(x|z) is something like the log of the mean squared error, then the density p(x|z) should be a squared error function, which is not even a valid distribution.

To be more precise, according to the authors, if one takes a Gaussian p(x|z), the reconstruction error should be modeled with

-log p(x|z) = log 1/N sum (x_i - \hat x_i)^2  + const 					(1)

where \hat x_i is a function of z. Note the logarithm on the right-hand side here. In appendix A, the authors observe that most other implementations instead optimize

-log p(x|z) = sum_{i=1}^N (x_i -\hat x_i)^2/(2\sigma^2) + N/2 log(2 pi sigma^2) 		(2)

Where often sigma is set to ½ (so that the last term on the lhs of (2) drops out when gradients are taken). The authors claim the latter is incorrect because sigma should be equal to the (biased) empirical variance. They then insert sigma=empirical variance into (2), take derivatives while ignoring the dependence of sigma on \hat x_i, and then arrive at something like eq. 1. They also argue that those implementations that use sigma=½ are actually optimizing a beta-VAE because of this “incorrect” prefactor of the reconstruction error.
I’m confident that the above claims are not correct.
As an example, using a Gaussian decoder distribution requires parameterizing the mean and variance of that Gaussian distribution as a function of z. Here, setting sigma = ½ (as is commonly done in other literature) is valid, contrary to the above claim. One is always allowed to just set the variance to a constant and ignore its dependence on z. This just leaves you with a less flexible distribution to model p(x|z).

The authors appear to use (1) as the reconstruction error term in the ELBO during VAE optimization, and then play with different prefactors of the KL term similar to beta-VAEs. The combination of the reconstruction error (1) and the prefactor no longer makes this a VAE, but more like a regularized auto-encoder with an unconventional reconstruction error that should not be interpreted as coming from the negative of a log density. Due to the logarithm in front of the mean squared error, the loss function is less sensitive to large errors in reconstruction. It is surprising that the reconstruction error does not cause overflow as log (0) --> - infinity.

Comments on experiments and related work:
- Experiments only show images of reconstructions of a VAE and LSR-GAN, a mean squared error plot of reconstruction errors and an accuracy plot of a classifier evaluated on the reconstructions produced by both models. In terms of MSE the LSR-GAN actually performs worse than their VAE baseline. The authors also train a classifier on cifar-10 and measure its accuracy using the ground truth labels and the reconstructed images of a VAE and LSR-GAN. Here the LSR-GAN performs marginally better than the VAE, but the overall accuracy is very poor.
- No experimental comparison is made against other VAE-GAN hybrids. Related work on hybrid VAE-GANS is discussed in the appendix, not in the main paper.

Other sections:
- Section 3 contains a very long interpretation of the minimum description length (MDL). It is unclear what the goal of this section is, as it does not lead to any results, nor does it help bring the point across as to why the proposed model is good at imagining the latent space of a VAE.


Additional feedback to improve the paper (not part of decision assessment):
The additional loss term of the generator is very similar to what is used in reweighted wake sleep, although this is not mentioned. It could be worth making a connection here.
In the conclusion the authors state “VAEs are often taken to be a pauper’s GAN’. This is not a very scientific statement and can be perceived as insulting for various reasons. Please rephrase this.


**Experience Assessment:**

I have published one or two papers in this area.

**Review Assessment: Checking Correctness Of Derivations And Theory:**

I carefully checked the derivations and theory.

**Review Assessment: Checking Correctness Of Experiments:**

I carefully checked the experiments.

**Review Assessment: Thoroughness In Paper Reading:**

I read the paper thoroughly.

---

> ### Author Response · Authors · 2019-11-08
> **Response to Reviewer #2**
>
> We strongly defend our claim about the form of the reconstruction error.
> Although this was made more as a side-remark we believe that it is very
> important to calculate the reconstruction error as we stated.
>
> The reconstruction error is actually an expected negative log
> probability, which by definition is the entropy of the error.  Under the
> standard assumption that the errors are zero mean normal variables this
> term should be the entropy of a Gaussian/normal distribution which is
> indeed equal to $\log(\sigma^2)/2+const$ as we assert.  The exponential
> of the negative entropy for a zero mean Gaussian is not a Gaussian
> distribution (or a distribution at all) so we don't accept your
> statement that this has to be wrong---it follow algebraically by putting
> in the definition of $\sigma^2$ into equation (2).  If you want to put
> in an unbiased statistical estimator for the variance it just changes
> the constant term by a very small amount.  Given we don't know $\sigma$
> a-priori using an empirical estimate seems the only alternative (given
> the number of errors we also expect that the empirical estimator (biased
> or unbiased) will be a very good approximation.
>
> We accept that our term is not common (we have not seen it used
> anywhere), but we are entirely confident that it is correct!
>
> The assertion that you can use any value of sigma you choose, we find
> difficult to understand.  If instead of using $\sigma^2=1/2$ we used
> $\sigma^2=1/20$ this would be equivalent to changing the relative
> proportion of the KL term by a factor of 10.  In the "true" VAE this
> should be the a true probability of making an error.  Given the errors
> have a variance, we should use that variance.  When you do this you will
> get the expression we derived.
>
> The reconstruction error will only overflow if the mean squared error
> for all pixels, colour channels and the whole mini-batch is zero.  We
> have never experienced this!  The modification is very easy to
> implement.  All you need to do is replace MSE in the loss function by
> N*log(MSE) where N is the number of colour channels times the number of
> pixels times the size of the minibatch.  In our experience this is
> equally easy to learn and will give you better reconstructions (although
> this could depend on the scaling you use in your implementation).
>
> Although in the MDL interpretation it is clear that the negative
> log-probability term is just an entropy which you wish to minimise, it
> is also the case in the original paper that you must minimise the
> log-probability of the errors.  Arbitrarily choosing $\sigma^2=1/2$ no
> longer makes this a probability of anything meaningful.
>
> *** Comments on experiments and related work
>
> There are no experimental comparisons because that was not the point of
> the paper.  The paper was about understanding VAEs.  LSR-GAN was
> introduced as a tool to visualise the information stored in the latent
> space of a traditional VAE.  Other VAE-GAN hybrids are not intended to
> do this so a direct comparison makes no sense.
>
> *** Other Sections
>
> The whole point of the paper is to explain what at VAE does using the
> MDL framework.  This, in our opinion, provides some new insights.
> Without Section 3 there is really no content.
>
> *** Additional Feedback
>
> Our comment "VAEs are often taken as a pauper's GAN" was to succinctly
> capture two aspect of VAE (a) they are easier to train than GANs and
> (b) they produce blurrier outputs than GANs.  However, the whole paper
> is written to point out that VAEs produce blurry reconstructions by
> design.  It is not the real point of VAEs.  VAEs produce rich latent
> embeddings of images.  Our intention was very far from insulting VAEs
> (clearly we would not have written the paper if we felt VAEs were
> uninteresting).  We can, of course, rephrase this, we had deliberately
> chosen the phrase to capture one view of VAEs, but the point of the
> paper is to show that this view is superficial and misses the point.

---

> > ### Comment · AnonReviewer2 · 2019-11-15
> > **Response to rebuttal**
> >
> > Thank you for the rebuttal.
> >
> > If the goal is not to propose a new model, but to provide new insight into the workings of a VAE, I fail to see what the actual new insights are that this paper provides.
> >
> > On the reconstruction error term:
> > - Your reconstruction loss ($N \log(E_{MSE})/2$) is unbounded from below, this cannot be a good loss. Stating you’ve never observed numerical instabilities due to this does not mean it’s a sound thing to do.
> > - If you model $p(x|z) = N(x|\mu=\hat x(z), \sigma^2(z))$, the exact choice of the dependence of $\sigma^2(z)$ on $z$ is a modeling choice for your decoding distribution. Choosing $\sigma^2=const$ is one of the valid choices. Note that even though for this choice no gradients will flow through $\log(2\pi\sigma^2)$, it does influence the ratio between the reconstruction term and the KL.  And yes, when restricting $\sigma^2$ to a constant, picking a different constant will lead to a different ratio of the gradient sizes coming from the reconstruction term and the KL divergence, but this is a consequence of choosing a different parameterization for the Gaussian decoding distribution. One could also choose $\sigma^2$ to be a function of $z$, and independently vary the beta in a $\beta$-vae.

---

### Official Review · AnonReviewer1 · 2019-10-23
**Official Blind Review #1**

**Rating:** 3

**Review:**

The paper proposes a new method for improving generative properties of VAE model. The idea is to train VAE in two stages: at first, train the vanilla VAE, then at the second stage freeze the encoder part and train the decoder part as a GAN generator with an additional regularizer which encourages cycle consistency in the latent space. Also the authors claim that other VAE-GAN hybrids which try to improve VAE model are “misguided” and poor samples and reconstructions of VAE are the consequence of minimum description length problem.

Concerns:
1) The main concern about this paper is the inaccuracy and very general statements without theoretical or empirical justification. For example, the authors claim that “the whole point of VAEs is to capture only compressible information and discard information specific to any particular image”. What is the definition of “only compressible information” or “information specific to any particular image”? Is there an experiment which can support this statement? Other examples of such general statements: “strength of a VAE is that it builds a model of the dataset that does not over-fit”, “the latent code does not contain enough information to do the reconstruction”, “VAEs are not broken and “fixing” them is actually likely to break them”.
2) Poor experiment comparisons with other baselines. The authors compare their method only with the vanilla VAE which is clearly insufficient. The authors claim that other VAE-GAN hybrids break the “strength of the VAE”. Could you please provide examples of problems and provide experiments where VAE-GAN baselines will be worse than VAE or the proposed method?
3) The paper structure is very confusing. The main part and experiments part are mixed. Therefore, it is hard to follow the text. The experiment setup is not clearly stated. For example, it is unclear for which dataset luminance-normalized Laplacian was computed.

Overall, the paper proposes a new method and  gives an alternative view on the VAE model. However, statements from the paper are very pretentious and are not rigorously proven. Also there are not empirical comparisons with other VAE-GAN hybrids. Therefore, I would suggest rejecting the current version.

--------------------------------------------------------
Update after author rebuttal

Thank you for your thoughtful response. However, I still think that the paper does not give new insights about VAE model and has poor experiment justifications of their statements. Considering MDL interpretation of VAE it is not new (see [1]). Therefore, the contribution of this paper is very limited.

After reviewing the other reviews and the author rebuttal, I do not change my original score.

[1] Xi Chen et al.,  Variational  Lossy  Autoencoder, 2016


**Experience Assessment:**

I have published in this field for several years.

**Review Assessment: Checking Correctness Of Derivations And Theory:**

I assessed the sensibility of the derivations and theory.

**Review Assessment: Checking Correctness Of Experiments:**

I assessed the sensibility of the experiments.

**Review Assessment: Thoroughness In Paper Reading:**

I read the paper at least twice and used my best judgement in assessing the paper.

---

> ### Author Response · Authors · 2019-11-08
> **Response to Reviewer #1**
>
> *** Concern 1)
>
> We believe the statements we have made are justified both theoretically
> and empirically.  We take the statements highlighted
>
> "the whole point of VAEs is to capture only compressible information and
> discard information specific to any particular image"
>
> This is a crucial argument we believe is justified by the minimum
> description length (MDL) interpretation of the VAE loss function.  The
> loss function consists of an expected negative log-probability or the
> entropy of the errors.  By Shannon's theorem this is also the minimum
> code length needed to communicate the error.  The KL term is a relative
> entropy that has an information theoretic interpretation of the code
> length needed to communicate a random variable with distribution $q(z|x)$
> using a coding with an underlying distribution of all code words of
> $p(z)$.  Therefore we can see the VAE loss function as the cost of
> communicating the original image by sending a code $z$ that the decoder
> decodes as $\hat{x}$ and an additional cost of repairing the image by
> communicating the errors.
>
> As a consequence, if a single image contains a butterfly in a corner, it
> makes no sense for this to be encoded in the latent space as this would
> be at least as costly as communicating the error.  If many images
> contained butterflies then you can reduce the overall code length by
> coding this into the latent space.  In contrast, an autoencoder would
> attempt to encode all information in the latent space.  This is what we
> mean by "only compressible information" and "information specific to a
> particular image".  We believe these are theoretically justifiable
> statements by the MDL interpretation.
>
> "strength of a VAE is that it builds a model of the dataset that does
> not over-fit""
>
> This follows theoretically from the above argument, but it leads to an
> interesting prediction.  A normal autoencoder (or a beta-VAE with
> beta<1) will provide better reconstructions of the training images (we
> didn't explicitly show this, but this is quite well known).  However, it
> won't produce better reconstructions on unseen testing images because the
> extra-information encoded by the autoencoder is image specific
> (over-fitting the training set).  The graphs in Figure 5 confirm this
> prediction empirically showing for a beta-VAE the reconstruction error
> on images from the CIFAR-10 test set are the same for $\beta\leq1$ and
> only become worse when $\beta>1$.
>
> "VAEs are not broken..."
>
> Our point is that VAEs are designed to balance the two terms in their
> loss function which will lead to poor reconstructions.  Fixing this
> would mean that image-specific information was encoded in the latent
> space which is precisely what VAEs are designed to avoid.
>
> *** Concern 2)
>
> As we explained this paper is about different view of VAEs.  LSR-GAN was
> not designed to produce better images than other GAN-VAE hybrids its
> purpose is to show (imagine) what is encoded in the latent space of a
> VAE.  Perhaps it was a mistake to introduce LSR-GAN as it has clearly
> been a distraction, but its purpose was to make a point that the latent
> space of a VAE has to be imagined as it does not encode all the
> information in an image.
>
> *** Concern 3)
>
> Clearly we have to accept that the paper has failed to communicate what
> we set out to communicate.  However, as we pointed out this is different
> to most papers.  Our new method is a visualisation tool to understand
> VAEs rather than yet another attempt to generate high quality images.
>
> *** General Comment
>
> We feel this is a bit harsh.  A lot of the argumentation seems to have
> been missed leading to an impression that our statements were rhetorical
> rather than based on a sound theoretical basis.  We chose a tone to
> highlight that we are challenging common held beliefs, but we can clearly
> change this if has caused confusion.

---

### Author Response · Authors · 2019-11-08
**To All Reviewers**

We would like to thank the reviewers for the time spent on reviewing the
paper.  The purpose of the paper is to take a step back and ask the
bigger question "What is the purpose of a VAE?" (why add the KL term).
We believe understanding this is important when trying to improve their
performance.  It is not our intention to propose a new network that
beats the current state-of-the-art on some metrics.  We propose LSR-GAN
as a tool for visualising what is encoded in a standard VAE (as we argue
this involves imaging details that won't be encoded).  As other VAE-GAN
hybrids were not designed for this purpose a direct comparison seems
redundant.

---

### Decision · Program_Chairs · 2019-12-19

**Decision:**

Reject

**Comment:**

The paper proposes a new method for improving generative properties of VAE model.  The reviewers unanimously agree that this paper is not ready to be published, particularly being concerned about the unclear objective and potentially misleading claims of the paper. Multiple reviewers pointed out about incorrect claims and statements without theoretical or empirical justification. The reviewers also mention that the paper does not provide new insights about VAE model as MDL interpretation of VAE it is not new.